# The Placement of Children in Need of Out-of-Home Care: Forms of Care and Differences in Attachment Security and Behavioral Problems in the Italian Context

**DOI:** 10.3390/ijerph20237111

**Published:** 2023-11-25

**Authors:** Rosalinda Cassibba, Caterina Balenzano, Fabiola Silletti, Gabrielle Coppola, Alessandro Costantini, Stefania Giorgio, Alessandro Taurino, Charissa S. L. Cheah, Pasquale Musso

**Affiliations:** 1Department of Educational Sciences, Psychology, Communication, University of Bari Aldo Moro, 70122 Bari, Italy; fabiola.silletti@uniba.it (F.S.); gabrielle.coppola@uniba.it (G.C.); stefania.giorgio@uniba.it (S.G.); alessandro.taurino@uniba.it (A.T.); pasquale.musso@uniba.it (P.M.); 2Interdepartmental Training and Research Centre for Care and Protection of Children and Families, University of Bari Aldo Moro, 70122 Bari, Italy; caterina.balenzano@uniba.it (C.B.); alessandro.costantini@uniba.it (A.C.); 3Department of Political Sciences, University of Bari Aldo Moro, 70122 Bari, Italy; 4Department of Psychology, University of Maryland, Baltimore County, Baltimore, MD 21250, USA; ccheah@umbc.edu

**Keywords:** adoption, foster care, institutional care, attachment, behavioral problems

## Abstract

The current paper investigated differences in secure attachment levels and behavioral problems among four groups of children in out-of-home care in Italy: closed adoption (child and birth parents not in contact following adoption), open adoption (child and birth parents still in contact after placement), foster care (child living temporarily with relatives or unrelated foster parents) and institutional care (child in residential care for large groups of children). One hundred and thirty children aged 10–19 were included in this study. The Attachment Interview for Childhood and Adolescence and the Achenbach Youth Self-Report were employed to measure participants’ secure attachment levels and behavioral problems. Both a multivariate analysis of covariance and measured variable path analysis were performed. Age, gender and time elapsed between the request for child protection and placement on out-of-home care were used as covariates. The results showed that adolescents in closed adoption had higher secure attachment scores than those in foster care and institutional care, while adolescents in open adoption scored significantly higher on problem behaviors than those in the other out-of-home care groups. Findings were discussed in terms of limitations and implications for future research.

## 1. Introduction

According to the literature, early traumatic experiences, such as neglect, emotional and psychological abuse, or physical and sexual abuse, as well as parental loss or witnessing domestic violence, are major risk factors for children’s socioemotional adjustment. Trauma may have a negative effect on attachment development [1] and may be associated with a higher risk of behavioral (i.e., internalizing and externalizing) problems [2,3].

Evidence converges to indicate that most children with out-of-home caregiving events have experienced at least one trauma in their short lives (see [4]). As a result of their often very traumatic history, children in out-of-home care represent a highly vulnerable category for attachment and mental health problems (e.g., [5]). However, removing the children from the problematic biological family and placing them in out-of-home care, whether temporary or permanent, may offer positive relational experiences, which might compensate for the preplacement adversities and promote their recovery (e.g., see [5,6]). For instance, placement in a family or community might allow the children to reorganize their own real and internalized relationship models thanks to the new relational experiences that the different reception contexts are able to offer.

Given that there are different forms of out-of-home care around the world, it would be beneficial to understand whether certain typologies might be more effective than others in terms of their influence on children’s secure attachment and the management of behavioral problems. To date, there has been little research on this topic. Furthermore, most previous works examining attachment and behavioral problems among out-of-home care individuals have focused on toddlers and preschoolers, and usually on one type of out-of-home care (see [7]).

As far as we know, there are no studies comparing both the levels of attachment security and the behavioral problems of older children and adolescents living in different forms of out-of-home care. From this point of view, Italy is a rather interesting context, as there are at least four different forms of out-of-home care: closed adoption, open adoption, foster care and institutional care (see below for the respective characteristics). In general, late childhood and adolescence are challenging developmental periods [8], but for those who are placed outside the birth home, it might be even more difficult [9]. Therefore, further studies on this topic are needed to better inform social workers and policy makers to promote the health development of youth in out-of-home care.

### 1.1. Forms of Out-of-Home Care in the Italian Context

As mentioned above, there are various out-of-home care services around the world. In Italy, four different forms principally occur: closed adoption, open adoption, foster care and institutional care [10,11,12,13,14,15]. Closed (or confidential) adoption involves no contact or interaction of any kind between child and birth parents after the adoption takes place. Open (or fully disclosed) adoption, instead, involves the maintenance of relationships between the child and the parents or other members of the birth family, even after the adoptive parents have assumed the full parental role (contact can be maintained through in-person meetings, telephone calls and the exchange of gifts or photos, for example). Foster care is a temporary substitute placement outside the child’s birth home (with a maximum duration of 24 months and some possible extensions), which usually takes place in a new family environment consisting of living with relatives (kinship care) or with nonrelative foster parents. Lastly, institutional care is a type of residential care for large groups of out-of-home children.

While sharing the aim of promoting the growth and adaptation of children, all these forms of out-of-home care partially pursue different specific objectives, relying on specific emotional, relational and contextual “resources”. With regard to closed adoption and foster care, they offer the child a privileged relational experience, with the presence of a family nucleus, where caregivers are more defined and constant persons compared to in institutional care; however, at the same time, they are two forms of care with very different temporal and child development goals. In fact, although both a closed adoption family and foster care family may offer a relational context, helping the child to build new emotional, cognitive and social skills, foster care families must constantly bear in mind that their involvement in care is temporary (since the ultimate goal is to reintegrate the child into the family of origin), while adoptive families are stably new families and, therefore, they probably tend to be more involved emotionally and relationally in the children, making them feel like a real “son/daughter” rather than a temporary passing child (e.g., see [16]). This is why “sometimes foster parents are warned not to commit to children placed in their care because of the inherent instability of foster care” and they may fail “to invest in a child for whom they care” (see [17], p. 517). Such a situation is particularly relevant in contexts such as that of Italy, where social workers prefer that children are removed from their home only when necessary and with the aim of reunification with the birth family; this results in concerns about the foster care process, probably due to the idea that it may not sufficiently preserve children’s relationships with their biological parents and relatives [18]. To overcome the problems associated with long-term foster care, which can lead to situations of fragility and vulnerability in foster children over time (for example, when they reach the age of 18, they no longer have legal protection), one possible solution is to transform foster care into open adoption, which, as mentioned above, aims to maintain children’s relational and emotional contact with their biological parents and family in the postadoption period after having experienced it during the foster care period. Open adoptions are often situations in which the children have not suffered extreme trauma of material or moral abandonment and, therefore, cannot be declared adoptable; in these circumstances, the biological parents are directly asked to give their consent to adoption outside the birth family. All these characteristics clearly distinguish open adoption from closed adoption (see [10]). The research results on open adoption are somewhat inconsistent. Some authors have reported that maintaining contact with the birth family positively influences the socioemotional adjustment of adoptees [13,19,20]. Other authors report that the quality of contact is the most important dimension to consider (see [10]), and that maintaining contact with a family of origin that is neglectful, emotionally abusive and violent could have detrimental rather than positive effects. In the latter case, it is plausible to expect that direct relationships with the biological family can limit the possibility of positive changes in the child’s relationship and behavioral patterns and foster “problem situations” that arise as a result of the child’s continued exposure to the negative role models of the biological parents.

Compared to the forms of out-of-home care described thus far, institutional care offers very different relational contexts, both because of the internal organization of the institution (e.g., the need for shifts and the interchangeability of some roles) and because of the goals pursued, such as promoting the autonomy of individuals approaching adulthood. Given these characteristics, previous works have shown that children in institutional care are at increased risk of attachment disorders and social and behavioral developmental delays, especially when staff-to-child ratios are low and the quality of care decreases due to staff turnover (for an overview, see [21]).

In further comparing the characteristics of these forms of out-of-home placement, it should be emphasized that families available for adoption, both closed and open, are subject to a much more demanding selective assessment of parenting skills and personality structures than families available for foster care. According to Italian laws 184/83 and 149/2001, parents who wish to adopt a child, after having submitted a declaration of willingness to the juvenile court, undergo various types of assessments. First, the court determines whether the couple meets the initial requirements (e.g., the age of the members of the couple and the fact that they are married) and then asks public social services to carry out a standardized psychosocial assessment, which examines various areas, such as parenting skills, health status, economic level and the quality of the family environment, in order to determine a judgment on suitability for adoption. The selection process for foster families is different. Although social workers and psychologists assess the parenting skills of foster parents, their recruitment in Italy does not follow national regulations or standard procedures, and the assessment criteria are quite heterogeneous and far less restrictive than those for adoption. In this scenario, it is unlikely that children with disabilities or special needs are placed in foster care. It is much more likely that they follow privileged adoption routes or are placed in institutional care [22]. However, apart from these characteristic legal and procedural aspects, it must be emphasized, as mentioned above, that the literature primarily shows a significant difference between the mental health of children in institutional care compared to other forms of out-of-home care. This is likely due to the fact that children in institutional care are placed later, have suffered more abuse and neglect and have experienced more domestic violence [11]. All these features, together with all the others described above for the different forms of out-of-home care, suggest that each of them may have different effects on children’s attachment and behavioral problems.

### 1.2. The Importance of Attachment and Behavioral Problems

The attachment construct can be understood as a set of relatively stable mental representations that are the result of the organization of current and past interaction experiences with one or more attachment figures. Such mental representations reveal the degree of security and trust that people can express in the most significant relationships that they experience [10,23,24,25]. Securely attached children have experienced caregivers who are stable and responsive to their needs. They are, therefore, able to express their feelings and explore their environment. In contrast, children’s experiences with less responsive caregivers are associated with insecure forms of attachment (i.e., avoidant, ambivalent and disorganized), which represent less optimal coping strategies [26] and may negatively impact their future health and well-being (see also [27]). Put differently, children’s poor attachments to their caregivers are related to different negative outcomes, like concurrent and later aggressions and delinquencies (see [28]), while a more secure attachment constitutes a protective factor against developmental risks and is associated with healthy socioemotional consequences (see [10]). However, starting from this traditional qualitative classification of secure and insecure attachments, some authors have begun to propose an assessment method based on a continuum of secure attachment (e.g., [29]) to overcome the concerns raised by the secure/insecure dichotomy, which led to a loss of between-subject variability, as well as within-subject variability over time (see [30]).

Behavioral problems are defined as acts that pose a considerable risk to one’s own or others’ health and safety and/or have a significant negative influence on one’s own or others’ quality of life [31]. They include internalizing and externalizing problems [32]. Internalizing problems are disturbances in mood or emotion, such as anxiety and depression [33], while externalizing problems take the form of aggression, antisocial behavior and impulsivity [34]. Notably, children’s behavior problems may lead to adverse developmental outcomes, including mental health disorders, loneliness, academic problems and criminal activity (see [35]; see also [34]).

Given their characteristics, both attachment and behavioral problems may have a long-term effect on children’s psychological, social and emotional development. This explains why it is critical to target these constructs when considering particularly vulnerable groups such as out-of-home care children.

### 1.3. Forms of Out-of-Home Care and Differences in Attachment Security and Behavioral Problems

The environment in which children are nurtured is likely to influence their attachment pattern and socioemotional adjustment (see [27]). Early adversity and negative experiences in out-of-home care children can put them at risk for attachment and behavioral problems ([5]). At the same time, the new relationships experienced by children in out-of-home care might be associated with more positive developmental paths (e.g., see [6]). Nevertheless, specific out-of-home care services may be differently related to children’s attachment and behavioral problems. However, to date, studies that have explored attachment and socioemotional and behavioral problems in different out-of-home care are extremely scant. In general, adoption may be considered a protective factor as adopted children fared far better (e.g., in terms of attachment disturbances and psychopathology) when compared with children who were institutionalized or later returned to their birth families (see [16,36]). Additionally, in his review, Tarren-Sweeney [5] reported that institutionalized children have less mental health problems than those in foster care and, among these, those in kinship care (e.g., vs. family-type foster care) have fewer problems. Additionally, Chartier and Blavier [11] reported that the sociopsychological health of Belgian institutionalized children was lower than that of those in foster families. The authors also underlined how out-of-home care children who stayed longer with their birth parents had, in general, lower psychological health. However, other authors reported differing findings. Quiroga and colleagues [27], for instance, in a Chilean sample, found that children in institutional care and foster care were more likely to develop insecure and/or disorganized attachment styles and more socioemotional and behavioral problems than children living with biological parents, while the two forms of out-of-home care were no different from each other. Overall, given the paucity and inconclusiveness of prior works, further research on this topic is needed, also extending it to other types of out-of-home care (e.g., open vs. closed adoption), age groups (i.e., adolescents) and cultural contexts (e.g., the Italian one).

### 1.4. The Current Study

Given the above premises, the current study aimed to investigate whether different out-of-home care services corresponded to different outcomes of older children’s and adolescents’ developments and adjustments, measured as security attachment levels and behavioral problems. Considering prior scarce and mixed findings, we assumed an exploratory perspective. However, in general, we expected that closed adoption and institutional care would represent, respectively, the most positive and the most negative type of care in terms of detecting higher levels of secure attachment and lower levels of behavioral problems. Given the various pre- and postadoption factors influencing adoptees’ outcomes (e.g., [10]; see also [5,37]), we controlled not only for age and sex (usual demographic variables), but also for the time of exposure to adversity, as measured with the proxy variable of time elapsed between the first report of the case to the authorities and the new child’s placement. A long time, indeed, may correspond to a greater period of exposure to adverse situations before being placed in a form of out-of-home care, or having experienced changes/failures in previous placements, and could, therefore, act as a potential confounding variable. The choice of such a measure was conceptually in line with the adverse childhood experiences (ACEs; [38]) model, which argues that the number/sum of adversities to which one is exposed to is more relevant than the type of adversity. We also took into account the time spent in the form of out-of-home care, considering only those cases that had been placed for a minimum period of at least two years; in fact, a stable care context may influence on how children and adolescents reorganize their internal relationship models and on the expression of behavioral problems, and we aimed to detect exactly this point. Moreover, we explored whether attachment could be associated with behavioral problems (e.g., [39]) and the moderating role of forms of out-of-home care in this relation. Although, to the best of our knowledge, no study had pursued a similar research question thus far, we expected closed adoption to represent the most favorable condition for a negative association between a higher secure attachment and behavioral problems, as a secure attachment is usually considered a protective factor against developmental risks in the context of closed adoptions (e.g., [40]).

## 2. Materials and Methods

### 2.1. Participants and Procedure

We collected data on 130 cases of out-of-home care children living in Apulia, Italy. These cases were identified through a collaborative research process with different local agencies (i.e., juvenile court, social and placement services, adoption and fostering associations and residential childcare communities). The inclusion criteria for selecting the cases of out-of-home care children were as follows: a minimum of two years living in one of the four forms of out-of-home care (i.e., closed, open adoption, foster care and institutional care) and a minimum of 10 years of age, because of the kind of measures that needed to be administered. As mentioned above, the choice to consider a minimum of two years of care placement was mainly linked to the fact that, since we measured participants’ attachment as well as their expression of problematic behaviors (see Section 2.2), we needed to consider a minimum time to guarantee their ability to reorganize their internal working models (IWM; [41]) and behavioral outcomes; in other words, we focused on assessing participants’ current attachments and behavioral functioning after two years of a stable and persistent placement in one of the different forms of care. During the initial phase of identification, we cataloged a total of more than 400 case files, and then, we causally extracted 50 cases for each of the four forms of out-of-home care. Following this, the adoptive or foster care families or the institutional care services were contacted to ask if there could be an agreement to include the children in the study. Researchers sent a letter that generally described the study, ensured confidentiality and requested to fill out the participation authorization form. We finally obtained 130 authorizations to participate, with a participation rate of 65%. The involved participants were adolescents aged 10–19 years (54.6% male; M_age_ = 15.14; SD = 1.87). They experienced different forms of out-of-home care: 28 adolescents experienced closed adoption (21.6%), 32 adolescents experienced open adoption (24.6%), 32 adolescents experienced foster care (24.6%) and 38 adolescents experienced institutional care (29.2%). Regarding the time between the first reporting by social services for child protection and the placement in out-of-home care, for 9.6% of the sample, it was less than one year, for 40.0%, it was between one year and three years, for 8.0%, it was between three years and five years and for 42.4%, it was more than five years. Table 1 shows the characteristics of the participants broken down in the forms of out-of-home care. No payment was offered for participation. All the procedures followed the ethical principles of the Italian Association of Psychology (see https://aipass.org/chi-siamo/#ethical-code, accessed on 30 June 2023) and was approved by the University’s ethics committee.

### 2.2. Materials

*Sociodemographics*. We initially derived information about the participants’ gender, age and time of exposure to adversity (using the proxy variable of time elapsed between the request for the child’s protection and placement in out-of-home care) through the case file review. These data were then directly verified by asking the participants for confirmation. We limited the collection of additional sociodemographic data due to two main reasons: (a) the agreements with the territorial agencies that supported the research envisaged collecting and/or using only these sociodemographic variables, both for privacy reasons and to contain further frustrations for the participants and their families or communities, and (b) in cases where we were able to access more details of the children’s past history, the information was collected using a nonstandardized approach and, as a result, did not contain similar information and was often not comparable.

*Attachment Interview for Childhood and Adolescence*. We assessed the participants’ attachment status by using the Attachment Interview for Childhood and Adolescence (AICA; [42,43]), a simplified version of the Adult Attachment Interview (AAI; see [44]). The AICA follows the AAI’s structure, but has a simplified linguistic formulation suitable for younger people. The AAI consists of a semistructured interview lasting almost an hour, and evaluates the organization of attachment representations according to the level of coherence of the autobiographical story. As a function of this, the content of the AAI is usually audio-recorded and, subsequently, transcribed verbatim. This content becomes important for assessing both the quality of past experiences with attachment figures through five specific experience scales (loving, rejecting, neglecting, involving and pushing to achieve) and the current state of mind through 12 additional scales (the idealization of the mother, idealization of the father, lack of recall, anger toward the mother, anger toward the father, derogation, metacognitive monitoring, passivity, unresolved loss, unresolved trauma, coherence of transcript and coherence of mind). State-of-mind scales are commonly used to classify respondents into one of the following attachment categories: (a) secure attachment; (b) insecure–dismissing attachment; (c) insecure–preoccupied attachment; (d) unresolved attachment. The AAI has shown excellent levels of reliability and validity (for a review, see [23,25]).

For this study, we used a different approach from the categorical one. We assessed attachment using a method for scoring the secure (versus insecure) attachment as a continuous variable, as suggested by Waters and colleagues ([45]; for its use in other studies, see [46,47,48]). For each participant, we used the Waters and colleagues’ ([45]) discriminant function equation to obtain a continuous security attachment score, taking into account the ratings (ranging from 1 to 9) obtained on the following five state-of-mind scales: idealization of the mother, idealization of the father, anger at the mother, derogation and coherence. Specifically, to obtain a single continuous security attachment score, we multiplied an individual’s raw score on each state-of-mind scale by the corresponding unstandardized partial discriminant weights and, then, we summed all these products along with the intercept constant (see [45] for the weights and intercept values). More positive scores indicated a higher secure attachment (the theoretical range being from −6.49 to 3.61). The measurement accuracy was assessed with a double-blind evaluation of 26 of the 130 interviews (20%) by two independent coders. The inter-rater agreement on the categorical classification was 85% (Cohen’s k = 0.79; *p* < 0.001).

*Youth Self-Report (YSR)*. We assessed behavioral problems based on the preceding 6 months using the Italian version of the YSR [49,50], a self-report instrument designed for older children and adolescents. The YSR includes 112 items, most of which can be categorized into 2 scales of internalizing and externalizing symptoms and in further 8 subscales: (a) the internalizing scale comprises somatic complaints (10 items, e.g., “I have nightmares”), withdrawn/depressed (8 items, e.g., “There is very little that I enjoy”), and anxious/depressed (13 items, e.g., “I cry a lot”) subscales; (b) the externalizing scale comprises aggressive behavior (17 items, e.g., “I argue a lot”) and rule-breaking behavior (15 items, e.g., “I don’t feel guilty after doing something I shouldn’t”) subscales; (c) the 3 remaining subscales of attention problems (9 items, e.g., “I have trouble concentrating or paying attention”), thought problems (12 items, e.g., “I can get my mind off certain thoughts”) and social problems (11 items, e.g., “I feel lonely”) define an “other problems” scale. Items were rated as 0, 1 or 2 (from *not true* to *very true or often true* to *very true or often true*). We obtained a total score by summing the related items; greater scores indicated higher levels of behavioral problems. As suggested by Achenbach and Rescorla ([49]), the raw total behavioral problem scores were standardized (t-scores) to compute the analyses (t-score = 60 representing the cut-off value for discriminating normal vs. clinical cases). The reliability and validity of the YSR were well established [49]. Cronbach’s alpha value for this study was 0.85.

### 2.3. Data Analysis

We followed three steps for the data analysis. First, we carried out descriptive statistics and univariate normality analyses for the key study variables, i.e., response percentages and/or means and standard deviations (SDs), the score range, skewness and kurtosis. We also checked for potential multivariate outliers by using the Mahalanobis distance and the Mardia’s multivariate kurtosis coefficient. Furthermore, Pearson’s correlations among the control and study variables were calculated.

Second, to assess whether different forms of out-of-home care resulted in different adolescent outcomes of security attachment levels and behavioral problems, we carried out a multivariate analysis of covariance (MANCOVA) using forms of out-of-home care as the predictor variable and the scores of the secure attachment and behavioral problems as the outcome variables. The gender, age and time elapsed between the request for the child’s protection and placement on out-of-home care were entered as covariates. We used the Statistical Package for the Social Sciences (SPSS) version 24 (IBM Corp., Armonk, NY, USA) to perform the analyses in these initial two steps.

Third, we carried out a measured variable path analysis (MVPA, [51]) within *Mplus 8.0* [52] to further evaluate how the forms of out-of-home care, as well as control variables, contributed to the participants’ secure attachment levels and behavioral problems (see Figure 1 for the theoretical model we tested). Additionally, using a multiple-group MVPA, we explored the moderating role of the forms of out-of-home care in the association of secure attachment levels with the behavioral problems (see Figure 2). To evaluate the model fit, we referred to the following goodness-of-fit indices [53]: chi-square test (χ^2^; *p* should be 0.05 or greater), comparative fit index (CFI; its value should be 0.95 or greater), Tucker–Lewis index (TLI; its value should be 0.95 or greater), root-mean-squared error of approximation (RMSEA; its value should be 0.05 or lower) and standardized root mean square residual (SRMR; its value should be 0.05 or lower). In comparing the nested models (i.e., more restrictive vs. less restrictive), we established that at least three out of the following four criteria had to be fulfilled to ascertain significant differences: χ^2^ with *p* < 0.05, ΔCFI ≤ −0.005, ΔRMSEA ≥ 0.010 and ΔSRMR ≥ 0.005 [54].

## 3. Results

### 3.1. Preliminary Analyses

As there were few missing values in the total sample (3.8%; two missing values on the secure attachment variable and three missing values on behavioral problems, not co-occurring with the former ones), we used the regression estimation function in the SPSS to impute them at the item level. This permitted us to fully use the data in the analyses. Before taking this choice, we also tested the use of the full information maximum likelihood (FIML) estimation method in the *Mplus* environment, which did not significantly change the final results and conclusions. Table 2 and Table 3 display descriptive statistics and normality analyses for the total sample, as well as within each form of out-of-home care. Specifically, Table 2 presents the percentages of participants who fit into the secure or insecure attachment category, showing how the insecure attachment category was clearly prevalent in the total sample as well as in the open adoption, foster care and institutional care subsamples, while in the closed adoption subsample, the secure attachment category evidently prevailed.

Table 3 reports on the means, SDs, score range, skewness and kurtosis for the key study variables. The skewness (<|1.21|) and kurtosis (<|1.52|) values fell in the −2 to +2 range, indicating univariate normality (see [55]). The multivariate normality analyses revealed no outliers. The secure attachment score was moderately high in the closed adoption group, slightly low in the total sample and in the open adoption group and moderately low in the foster care and institutional care groups. The behavioral problem scores were, on average, below the threshold for clinical relevance (as mentioned, equal to 60) in both the total sample and all the subsamples, although the open adoption subsample came close to that threshold and seemed to score somewhat higher than the other subsamples. Table 4 shows correlations among the control and study variables. No significant associations were revealed within the entire group. Within the forms of out-of-home care, gender (0 = male; 1 = female) was linked to the secure attachment level in the foster care subsample (girls showing higher levels than boys), while age was positively associated with (a) behavioral problems in the closed adoption subsample, (b) time between request for the child’s protection and placement in out-of-home care in the foster care subsample and (c) secure attachment level in the institutional care subsample. No significant links were evidenced between the secure attachment level and behavioral problems.

### 3.2. Differences between Out-of-Home Care Subsamples in Secure Attachment Levels and Behavioral Problems

The MANCOVA on the secure attachment levels and behavioral problems showed a significant multivariate effect of the forms of out-of-home care, with Wilks’ Lambda = 0.84, F(6, 244) = 3.82, *p* < 0.01, η^2^ = 0.09, after controlling for gender, age and time elapsed between the request for the child’s protection and placement in out-of-home care. Table 5 summarizes the follow-up analyses. The results indicated significant score differences among the forms of out-of-home care for both the secure attachment level and behavioral problems. Specifically, the pairwise comparisons (*p* < 0.05) showed that participants in closed adoption care scored higher on secure attachment compared to those in institutional care, but no significant differences were evidenced between these two forms of out-of-home care and both open adoption care and foster care. Additionally, open adopted adolescents scored higher on behavioral problems than adolescents in the other forms of out-of-home care.

### 3.3. Associations of Forms of Out-of-Home Care with Secure Attachment and Behavioral Problems

We further explored our research question by using a MVPA approach with a multicategorical predictor (forms of out-of-home care) directly affecting the two observed variables of secure attachment and behavioral problems. Control variables (gender, age and time elapsed between the request for the child’s protection and placement in out-of-home care) were permitted to predict the outcome variables as well as to covariate with the forms of out-of-home care and to each other, except for the covariation between age and gender, which was constrained to zero to allow for the model estimation (as suggested through bivariate correlations, see Table 4, r was < 0.10). To represent the four groups of out-of-home care in the model, a dummy coding approach was pursued (see [56]). Table 6 describes the indicator coding system. The closed adoption category functioned as the reference group and was not coded explicitly, while the other categories (the dummy variables) presented the value of 0 for the cases in the reference group. Thus, estimated parameters were interpreted relative to the closed adoption category. We also replicated the analysis using the other forms of out-of-home care as reference groups. To test the model, we used the maximum likelihood estimation method.

The estimated model had excellent fit, χ^2^(1) = 0.55, *p* = 0.46, CFI = 1.00, TLI = 1.31, RMSEA = 0.000, SRMR = 0.011. The results significantly showed that the dummy variables of foster care vs. closed adoption and institutional care vs. closed adoption were negatively associated with secure attachment scores (see Figure 3a), meaning that adolescents in foster care and institutional care experienced less secure attachments relative to the adolescents in closed adoption (and vice versa; see Figure 3c,d). Additionally, the dummy variable institutional care vs. open adoption was negatively associated with secure attachment scores (see Figure 3b), that is, adolescents in institutional care experienced less secure attachment relative to the adolescents in open adoption (and vice versa; see Figure 3d). Adolescents in closed adoption presented secure attachment scores not significantly different from those in open adoption (see Figure 3a,b), and adolescents in foster care presented secure attachment scores not significantly different from those in both open adoption and institutional care (see Figure 3b–d).

The results also showed that adolescents in open adoption experienced significantly higher behavioral problems relative to adolescents in the other forms of out-of-home care. Furthermore, no significant associations were evidenced (a) between control variables and secure attachment and behavioral problems, and (b) between secure attachment and behavioral problems.

By considering the forms of out-of-home care as a moderating variable of the association between secure attachment scores and behavioral problems, we performed a multiple-group MVPA. We used the same previous model, except that the multicategorical predictor variable became a grouping variable useful for comparing the associations between secure attachment levels and behavioral problems in the four forms of out-of-home care. The initial unconstrained model (no equality constraints imposed for the association of interest across the four forms of out-of-home care) had a very good fit, χ^2^(4) = 3.74, *p* = 0.44, CFI = 1.00, TLI = 1.10, RMSEA = 0.000, SRMR = 0.045. The constrained version of this model (association between secure attachment levels and behavioral problems constrained to be equal across the four forms of out-of-home care) had a significantly poorer fit, χ^2^(7) = 13.28, *p* = 0.06, CFI = 0.664, TLI = −0.34, RMSEA = 0.166, SRMR = 0.067, Δχ^2^(3) = 9.54, *p* < 0.05, ΔCFI = −0.336, ΔRMSEA = 0.166, ΔSRMR = 0.022. The modification indices recommended releasing the constraint for the closed adoption and the institutional care groups. This partially constrained model had an adequate fit, comparable to the initial model, χ^2^(5) = 3.83, *p* = 0.57, CFI = 1.00, TLI = 1.35, RMSEA = 0.000, SRMR = 0.045, Δχ^2^(1) = 0.09, *p* = 0.76, ΔCFI = 0.000, ΔRMSEA = 0.000, ΔSRMR = 0.000. Figure 4 displays the standardized coefficients for this better model. The results evidenced that for the closed adoption group, higher secure attachment scores were associated with fewer behavioral problems, while for the institutional care group, the secure attachment scores were associated with higher behavioral problems. No significant association was revealed for the open adoption and foster care groups.

## 4. Discussion

This study aimed to investigate differences in secure attachment levels and behavioral problems between four Italian groups of children placed for at least two years in out-of-home care, that is, closed adoption (birth parents/child not in contact following adoption), open adoption (birth families/child still in contact after placement), foster care (living temporarily with relatives or unrelated foster parents) and institutional care (residence for large groups of children). It contributed by expanding previous research in at least three main respects. First, to our knowledge, no other study at the same time considered these four forms of out-of-home care. Second, most of the studies in this research area focused above all on children and less on adolescents, as this study did. Third, the research topic is still limited in the Italian context because previous research about children in need of out-of-home care focused mainly either on the attachment or behavioral problems, but not simultaneously on both.

We attempted to answer our research question by recruiting participants to be directly involved in data collection. After controlling for age, gender and time elapsed between the first report of the case to the authorities and the new child’s placement (a proxy indicator for exposure to adversity), the results suggested that participants especially in the closed adoption group appeared to show the best adjustment outcomes. At the time of data collection, they experienced the highest secure attachment levels along with peers in the open adoption group, but participants in the latter group also exhibited the highest levels of behavioral problems. Participants in institutional care showed the lowest secure attachment levels, while those in foster care showed secure attachment levels and behavioral problems that were not significantly different from both peers in closed adoption and in institutional care.

Focusing on the secure attachment levels, the results showed quite consistently that the closed adoption group had the highest secure attachment levels, whereas the institutional care group had the lowest secure attachment levels. The open adoption group approximated the secure attachment levels of the closed adoption group. These findings corresponded to our expectations and previous works (see [16,36]). Although our findings did not allow for specific interpretations, in a speculative vein, we propose two potential interacting explanations. First, it may be that there was a selection bias, whereby older children and adolescents who show greater difficulties in the quality of relationships are more likely to be placed in an institutional care context, while those with fewer difficulties are more likely to enter the closed adoption circuit. Second, representations of lower secure attachments may be explained not only due to the first and unfavorable experiences within the biological family (events so harmful that the child must be removed), but also due to the failed reorganization towards a model of security in the care context, as it can more easily happen in institutional care. Indeed, it may be that institutionalized children lack a clear feeling of belonging and continue to feel like “waiting children”, which causes difficulty for them to have the opportunity to build contextualized secure attachment bonds. In the opposite direction, the adoption context, especially in the form of closed adoption, may represent a placement in which a positive reconsideration of secure attachment representations is more likely (this is greatly favored, at least in Italy, in preadoptive suitability assessments of future adoptive parents). This second explanation may be further supported by considering that care arrangements vary greatly in relation to the quality of supervision and time devoted to the child. In an institution, care providers split their time across a large group of youth and are not always present. On the other hand, during adoption, caregivers are much more present, allowing their children to have more support and more stable attachment figures. Hence, our study aligned with a scenario of ideas according to which the relationships that out-of-home care children build with new caregivers may differently interact with the attachment patterns formed in their previous history depending on the forms of out-of-home care [27], closed adoption being the context with greater potential for relational resources.

In regard to behavioral problems, adolescents in the open adoption scored significantly higher in problem behaviors than participants in the other out-of-home care groups. Speculatively, this result may suggest that remaining in contact with birth families (which is a feature of open adoption) may not always be beneficial for these children (see [10]). Continued contact with problematic and dysfunctional family contexts, in which parents are abusive and/or neglectful, could generate emotional malaise in the children, and, thus, internalizing behaviors, such as anxiety and depression. Furthermore, because these parents serve as negative role models, children may assume disadvantageous values and actions of them and maintain or develop more externalizing behaviors.

A further deepening of our analyses showed that a higher secure attachment seemed to be related to fewer behavioral problems in the case of closed adoption. Such results echoed prior works showing that, in adoptive contexts, a secure attachment may be a protective factor against developmental risks [40] and is associated with healthy socioemotional outcomes [57,58]. By contrast, children with lower secure attachment levels tend to exhibit higher behavioral problems, such as anxiety and depression (e.g., [39]). Thus, our findings were in line with the idea that closed adoption may be a form of out-of-home care that positively interacts with correlates of adoptees’ adaptive functioning. Finally, a surprising result was linked to the positive association between secure attachment levels and behavioral problems in the institutional care group. However, it must be noted that the secure attachment level was quite low for this group. A possible explanatory hypothesis was that it may be necessary to achieve at least moderate secure attachment levels before the latter can manifest optimal associations with socioemotional and behavioral outcomes (e.g., at least nonpositive links with behavioral problems).

## 5. Limitations and Conclusions

This study had limitations that needed to be noted. First, the sample size was small, and this could have affected our results, especially those regarding the multigroup MVPA. Although it is always difficult to have a high number of participants in this research area, future studies should still aim for larger samples. This would also allow for the use of more complex analysis models.

Second, we were unable to collect baseline measures regarding adolescents’ preadoptive attachment levels and mental health statuses. This and other information would have been particularly useful to highlight any selection bias that could lead specific children into certain care arrangements. For example, we might expect behavioral problems to be more frequent in young people going into institutional settings, because it has been difficult to place them in other forms of out-of-home care also due to their behavior. Additionally, it could be that some preferences of adoptive parents may influence their final decision to adopt a certain child. Although in Italy the risk of selection bias seems reduced, given that the decision on the form of care placement is usually taken by a judge in connection with the designated services considering the best interests of the minor, it seems appropriate that international research appropriately takes this issue into account.

Third, we measured the children’s secure attachment levels, but we were unable to collect data on caregivers’ characteristics to insert into our models. Given that children develop attachments in an interactive process with one or more caregivers, future research could also consider measuring some of these characteristics to understand how they are associated with the secure attachment levels of the children in their care.

Fourth, the cross-sectional design prevented us from illustrating how attachment levels and behavioral problems changed from the date of placement to the date of data collection, as well as from drawing clear conclusions about effects or causal links. Hence, further research is warranted on this topic. For example, longitudinal studies could better explain whether participants who show adequate secure attachment levels in a certain care setting are young people who can be defined as simply “secure” or having “earned” a secure attachment ([59]; see below for more details on this hypothesis).

Fifth, our impossibility of collecting information regarding children’s placement history resulted in a difficulty to “contextualize” our data and results with respect to possible situations of placement stability or instability. For example, adopted children may have spent time in institutional care before being adopted. Although we tried to control the related variability on this through the use of an inclusion criterion that envisaged a stability of at least two years in the form of out-of-home care (which, as previously mentioned above, simultaneously guaranteed a theoretically useful time for the reorganization of children’s internal working models), future research must be able to take into account how possible transitions into different forms of care may influence children’s outcomes at a certain point in their development path. Additionally, future studies should include direct measures of children’s exposure to adversity, given that the use of a proxy variable measuring a presumable time of exposure to adversity may be questionable (in our case, however, it was the most objective and comparable variable that we were able to obtain from the available information).

Despite these limitations, our study constitutes an interesting contribution to the field, as, to our knowledge, it is one of the first studies describing adolescents’ security attachment levels and behavioral problems in four forms of out-of-home care. Our findings may be useful as background for future research questions and hypothesis generation. Here, we outlined some of these questions and hypotheses. Although our results suggested that the group of participants who experienced closed adoption for at least two years tended to be those with the highest secure attachment levels, the question remains unanswered whether this condition was already present in the preadoptive phase or whether it improved over time. In other words, as also mentioned above, one could ask whether children and adolescents who already show a less problematic level of attachment are more likely to be included in closed adoption processes than in other forms of care, or whether adoptive parents represent decisive promoters of the development of their adopted children’s secure attachment levels, especially when initially low. Additionally, given that the manifestations of behavioral problems may change with age, understanding how age moderates children’s adjustment outcomes in the four forms of out-of-home care could be interesting and significant for choosing their placement. Furthermore, the different care contexts are often not uniform. Therefore, considering factors associated with the internal variability of each form of out-of-home care could be important to maximize adaptive regulation processes between children requiring out-of-home care and the specific care context. These and other issues open up research and study scenarios that are not only stimulating for academics and professionals in the area, but that also appear extremely necessary to guarantee the best paths to the resilience and/or well-being of children in need of out-of-home care.

## Figures and Tables

**Figure 1 ijerph-20-07111-f001:**
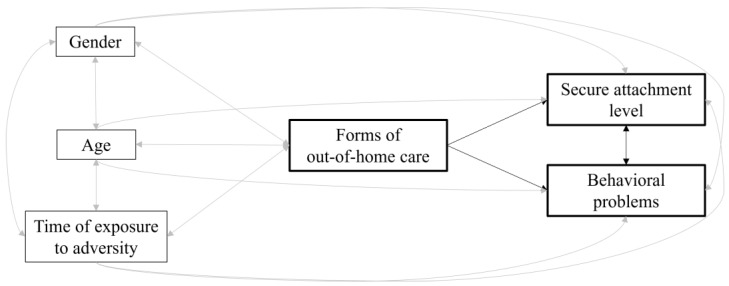
Theoretical model of the relations of the forms of out-of-home care with secure attachment levels and behavioral problems. The key study variables and their related associations are presented in black. Control variables and their related associations are presented in gray.

**Figure 2 ijerph-20-07111-f002:**
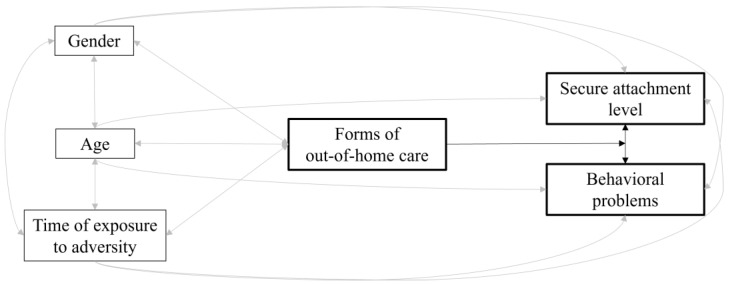
Theoretical model highlighting the moderating role of the forms of out-of-home care in the association of secure attachment levels with the behavioral problems. The key study variables and their related associations are presented in black. Control variables and their related associations are presented in gray.

**Figure 3 ijerph-20-07111-f003:**
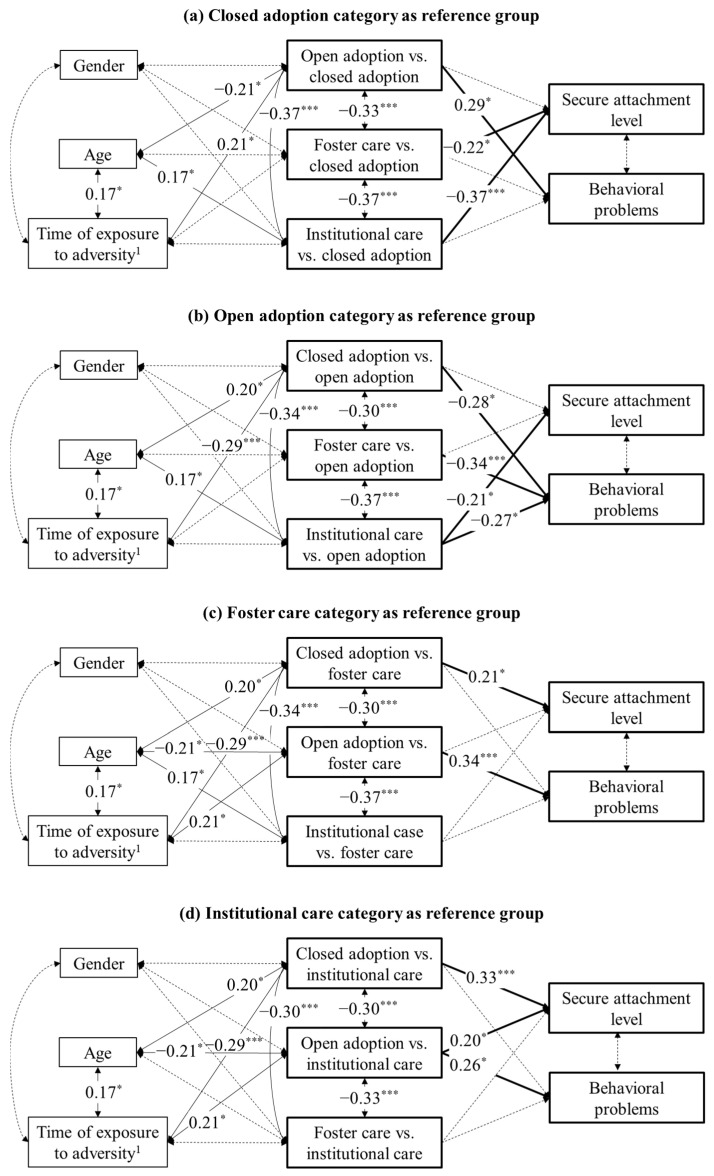
Results from the measured variable path analysis model based on maximum likelihood estimation. Standardized coefficients are presented. The key study variables and the associated significant pathways are shown in bold. Solid lines represent significant associations, dashed lines denote nonsignificant links. For better visualization, pathways (all nonsignificant) from control variables to the variables of secure attachment level and behavioral problems are not shown * *p* < 0.05; ** *p* < 0.01; *** *p* < 0.001. ^1^ As measured through the proxy variable of time elapsed between request for child’s protection and placement in out-of-home care.

**Figure 4 ijerph-20-07111-f004:**
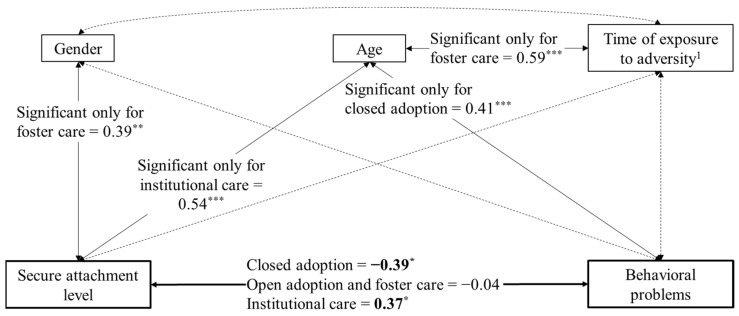
Results from the multiple-group measured variable path analysis model based on maximum likelihood estimation. Standardized coefficients are presented. Coefficients in bold are significantly different from the others when considering the same association. Solid lines mean that at least one coefficient in one group was significant. Dashed lines represent nonsignificant associations between variables. Groups are labeled with the respective out-of-home care. * *p* < 0.05; ** *p* < 0.01; *** *p* < 0.001. ^1^ As measured through the proxy variable of time elapsed between request for child’s protection and placement in out-of-home care.

**Table 1 ijerph-20-07111-t001:** Characteristics of participants broken down by the forms of out-of-home care.

Descriptive Variable	Closed Adoption	Open Adoption	Foster Care	Institutional Care
Gender (N (%))				
Male	13 (46.4%)	18 (56.2%)	18 (56.2%)	22 (57.9%)
Female	15 (53.6%)	14 (43.8%)	14 (43.8%)	16 (42.1%)
Age in years (M (SD))	15.86 (1.53)	14.47 (1.87)	14.59 (2.26)	15.63 (1.40)
Range	14–18	12–18	10–19	14–18
Time in years of exposure to adversity ^1^ (M (SD))	2.56 (1.57)	4.48 (2.22)	3.92 (2.23)	3.78 (1.96)
Range	0.5 to 5.1	1.5 to 6.2	0.9 to 6.0	0.7 to 5.9

^1^ As measured using the proxy variable of time elapsed between request for child protection and placement in out-of-home care.

**Table 2 ijerph-20-07111-t002:** Percentages of participants who fit into the secure or insecure attachment category.

Percentages of Secure and Insecure Attachment	Total Sample	Closed Adoption Subsample	Open Adoption Subsample	Foster Care Subsample	Institutional Care Subsample
Secure attachment	37.2%	67.9%	37.5%	31.2%	18.9%
Insecure attachment	62.8%	32.1%	62.5%	68.8%	81.1%

**Table 3 ijerph-20-07111-t003:** Means, standard deviations, skewness and kurtosis for the key study variables.

Observed Variable	Mean	Standard Deviation	Range	Skewness	Kurtosis
Total sample					
Secure attachment level	−0.32	1.66	−4.02 to 3.09	0.32	−1.17
Behavioral problems	52.43	10.94	24 to 79	−0.025	0.04
Closed adoption subsample					
Secure attachment level	0.63	1.83	−4.02 to 3.09	−0.89	−0.07
Behavioral problems	50.74	7.21	33 to 66	−0.33	0.15
Open adoption subsample					
Secure attachment level	−0.31	1.61	−2.28 to 2.40	0.46	−1.52
Behavioral problems	57.94	9.82	36 to 79	−0.03	0.41
Foster care subsample					
Secure attachment level	−0.53	1.50	−2.89 to 1.99	0.30	−1.28
Behavioral problems	49.29	13.03	26 to 75	0.08	−0.64
Institutional care subsample					
Secure attachment level	−0.85	1.46	−3.01 to 2.97	1.21	0.64
Behavioral problems	51.66	10.92	24 to 67	−0.62	−0.16

**Table 4 ijerph-20-07111-t004:** Pearson bivariate correlations (r) among control and key study variables.

	1	2	3	4	5
Total sample					
1. Gender (0 = male; 1 = female)	1				
2. Age	0.07	1			
3. Time of exposure to adversity ^1^	−0.13	0.16	1		
4. Secure attachment level	0.08	0.16	−0.13	1	
5. Behavioral problems	0.01	0.04	0.12	−0.01	1
Closed adoption subsample					
1. Gender (0 = male; 1 = female)	1				
2. Age	−0.14	1			
3. Time of exposure to adversity ^1^	−0.10	0.15	1		
4. Secure attachment level	−0.03	0.17	−0.27	1	
5. Behavioral problems	−0.02	0.38 *	−0.05	−0.21	1
Open adoption subsample					
1. Gender (0 = male; 1 = female)	1				
2. Age	−0.05	1			
3. Time of exposure to adversity ^1^	−0.25	0.14	1		
4. Secure attachment level	−0.05	0.12	−0.02	1	
5. Behavioral problems	0.06	0.21	0.27	0.02	1
Foster care subsample					
1. Gender (0 = male; 1 = female)	1				
2. Age	0.10	1			
3. Time of exposure to adversity ^1^	−0.01	0.58 ***	1		
4. Secure attachment level	0.36 *	−0.07	0.11	1	
5. Behavioral problems	0.10	0.17	0.06	−0.06	1
Institutional care subsample					
1. Gender (0 = male; 1 = female)	1				
2. Age	0.27	1			
3. Time of exposure to adversity ^1^	−0.11	0.04	1		
4. Secure attachment level	−0.03	0.48 **	−0.15	1	
5. Behavioral problems	−0.08	−0.30	−0.04	0.13	1

Note. * *p* < 0.05; ** *p* < 0.01; *** *p* < 0.001. ^1^ As measured with the proxy variable of time elapsed between request for child’s protection and placement in out-of-home care.

**Table 5 ijerph-20-07111-t005:** Univariate analyses of covariance and pairwise comparisons for the four forms of out-of-home care on the adolescent secure attachment levels and behavioral problems.

	MANOVA-Adjusted Means by Forms of Out-of-Home Care					
	Closed Adoption	Open Adoption	Foster Care	Institutional Care	F(3, 123)	η^2^
Secure attachment level	0.41 ^a^	−0.15 ^ab^	−0.42 ^ab^	−0.91 ^b^	3.82 *	0.09
Behavioral problems	50.71 ^a^	58.04 ^b^	49.48 ^a^	51.44 ^a^	3.86 *	0.09

Note. Within each row, scores that did not share any superscripts (a or b) in common among them differed significantly (*p* < 0.05). * *p* < 0.05.

**Table 6 ijerph-20-07111-t006:** Indicator coding system for forms of out-of-home care (closed adoption category as reference group).

	Forms of Out-of-Home Care			
Dummy Variables	Closed Adoption	Open Adoption	Foster Care	Institutional Care
Open adoption vs. closed adoption	0	1	0	0
Foster care vs. closed adoption	0	0	1	0
Institutional care vs. closed adoption	0	0	0	1

Note. The closed adoption category was not coded explicitly representing the reference group.

## Data Availability

The data presented in this study are available on request from the corresponding author. The data are not publicly available due to the specific agreements with the territorial agencies that supported the research.

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
