# Peer review of "The Placement of Children in Need of Out-of-Home Care: Forms of Care and Differences in Attachment Security and Behavioral Problems in the Italian Context"

_ijerph, 2023, doi:10.3390/ijerph20237111_

Round 1

Reviewer 1 Report

Comments and Suggestions for Authors

This research compares secure attachment levels and behavioral problems in adolescents who have been adopted in four modalities (closed adoption, open adoption, foster care, and institutional care).

I believe that the topic is very relevant and well justified, and they have also been very careful in their method and processes to protect the participants,

General suggestions

·      The authors do not mention the rating ranges that can be obtained by participants in attachment (lines 276-277) and I think this is important when looking at the means of these scores by group. In fact, I am left in doubt whether this score is categorical or a continuous variable, because if it is categorical, it does not make much sense to compare their means (table 3).

·      In line 347 there is a punctuation error: "Table 4. shows the correlations".

·      Revise the title bar of table 4, it has punctuation errors.

·      In table 5, despite the superscripts, the combination is not understood.

·      It seems to me that the authors repeat the findings information found from line 394 to line 405; in paragraphs 405 to 410 and could be summarized.

·      I suggest review the writing of the paragraph in lines 449-451, sorry I don't understand it.

·      Review the wording of line 506.

·      Review the writing of this sentence: “This suggests that a certain level of secure attachment 511 probably needs to be achieved for it to have optimal associations with socio-emotional 512 and behavioral outcomes” (lines 511-513).

Comments on the Quality of English Language

Minimum English proofreading required.

Reviewer 2 Report

Comments and Suggestions for Authors

This paper explores secure attachment and behavioral problems of young people based on the type of living arrangement.

The paper is well-presented and well-written.  My concerns, however, have to do with what the authors themselves identify as a limitation of the study.  As they note, they do not have longitudinal measures of attachment so they cannot say how attachment has changed from the date of placement to the present.  I would argue that it runs much deeper than that.  The authors do not raise the selection bias that leads children into certain care arrangements.  For example, prospective adoptive parents, whether closed or open, have preferences that likely affect their decision-making given the adoptable children who are available.  Children are not placed at random into institutional settings - behavior problems are to be expected in the population of young people who go into institutional settings.  That brings along with it whatever other correlates are associated with the behavioral problems.  I am afraid that these are serious limitations with implications for what can be said.

In addition, there are substantial issues with omitted variable bias.  For instance, the authors rightly identify caregiver investment as a feature of caregiver/child relationships but have no caregiver characteristics in their model.  Attachment is an interaction with a caregiver, not something the child develops independently of the response from a caregiver.

Similarly, there is little done with placement history - did the adopted children have prior foster care experience?  Was that pattern of care stable?  Unstable?  More generally, adopted children are at a different point along the service trajectory - could it be that they spent time in institutional care before being adopted?  Did that time in institutional care affect the chances of being adopted?  I see the sample had to have been in the setting for a minimum of two years - but there is no context given for how that compares on average.  For example, institutional care is often quite short - it isn't a long-term care situation.  If that is the case in Italy, what does it say about a young person who stays in institutional care for 2 years?  Are they at all representative of all children placed in institutional care?  How long does a typical stay in foster care last?  How are behavior problems and attachment correlated with time in care?  What can be said about children who are in care for less than 2 years - what fraction of the children placed are in care for less than 2 years?  In other words, the sample selection is problematic because of the correlations with the probability of selection into study criteria - a minimum of two years in one of the four forms of care.

I do note that the age range is broader among the open adoption cases, with younger children overall (i.e., the range) although with a wider standard deviation.  How long ago was the 19-year-old adopted as compared to the 10-year-old?  Age was significant only for closed adoption (Figure 2) - how much of this is due to the wider range?

I also question whether the time between a report and placement is indicative of exposure to adversity.  What was the basis for the report - neglect or abuse?  Exposure to adversity is not a binary experience - it happens in temporal proximity to the report or not.  There is a discussion of whether the report is a first report, prior placements, etc. but no details of how this was taken into account.  The time between report, allegation, and placement are likely correlated.  So, is whatever relationship observed a function of the timing or the allegation?  Does the allegation signify anything with regard to the type of adversity?

Also, the characterization of placement in a 'welcoming structure or family' represents the unmeasured characteristics of the caregiver - see comment above.  I appreciate the positive view of foster carers but their capabilities as an emotional context for caregiving are not so uniform.  Moreover, other than as a criterion for sample selection, shouldn't there be a control for how much time the young person spent in the caregiving arrangement?  Isn't this also an exposure variable?  If the caregiving context has an influence, then time spent in this or that situation is material to the discussion.  How long are institutional placements in Italy?  In Italy do adoptions disrupt?  If so, the only adoptions being evaluated are adoptions that persist.

Statistically, the separate models are really about interaction effects - the relationship between the independent and dependent variables is related to the level of a third variable.  I think the separate models are a bit much - overly complicated given the data at hand.  Also, if I am reading the models correctly, gender, age, and time of exposure do not have direct effects on either behavior problems or attachment.  If so, what is the justification for that?

Substantively, interaction effects with age would in my opinion be far more interesting:  among children of the same age, how do they compare from an attachment perspective, a behavior problem perspective?  Insofar as the manifestation of behavior problems changes with age, this perspective is missed with the simple controls for age.  

When it's all added up, the main issue is the strong language used to describe relationships between the independent and dependent variables.  To the extent the language sounds causal, the language goes well beyond what the authors are able to conclude, given all that is missing from the analysis.  For example:

"By contrast, an adoption process, especially in the form of closed adoption, seems to be the form of out-of-home care more capable of promoting secure representations of attachment among children and adolescents."

"More capable of promoting' is the sort of inference that isn't supported by the analysis.   It certainly seems plausible that securely attached children are the children who find themselves at some point in a closed adoption.  Said differently, are securely attached children the children most likely to be selected by prospective adoptive parents interested in closed adoptions.  It is certainly possible as an alternative explanation to what the authors have uncovered. If so the question becomes, given the sense of attachment when the adoption takes place, do the caregivers in closed adoptions sustain the level of attachment or promote the development of attachment in cases where the level is low to begin with?  The authors don't have this data, so it would be wise to avoid inferences that suggest otherwise.

There are questions raised about behavioral problems, etc. The argument about open adoptions and the negative consequence of contact with dysfunctional families is logical but not supported by the data.

Another example:

" . . . our findings endorse the idea that adoptive parents’ sensitivity is a key factor in preventing the emergence of behavioral problems."

I don't see how the "endorsement" works because the authors did not measure the quality of care.  

As the authors note, causal relationships are very difficult to ascertain from cross-sectional data.  For that reason, I do not think the study offers guidance for clinicians and social workers interested in supporting a young person's adjustment.  As for the policy implications and the preference for closed adoptions, I am afraid that is a bridge too far given the evidence.  It is, in fact, a bit frustrating when on the one hand the authors admit that their design limits the ability to draw causal inferences and on the other, they make policy recommendations with the idea that implementing the policy will have the intended causal effects.  You cannot have it both ways.

To be published, the authors have to contend with what they have in the way of data - a study that describes the secure attachment levels across care types.  I think a paper that recognizes what the authors have and don't have in the way of data would be publishable provided any hint of a causal model is set to the side until they have the necessary data.  

As an aside and not the authors' responsibility - the formatting of the tables makes them very difficult to follow in large part because the text in the first column of the table is centered rather than left justified.  The awkward page breaks and awkward line breaks don't help either.

Reviewer 3 Report

Comments and Suggestions for Authors

This paper examines differences in secure attachment levels and behavioral problems across children in four groups: closed adoption (birth parents/child not in contact following adoption), open adoption (birth families/child still in contact after placement), foster care (living temporarily with relatives or unrelated foster parents), and institutional care (residence for large groups of children).

Introduction

Please provide the directional hypotheses for each model and group.

Method

Please update your analytic approach measured variable path analysis (MVPA) because there is no latent variable in your hypothesized models. (See Mueller & Hancock, 2018).

The author(s) did not provide information about the sample size estimation. It is recommended to have 100 cases/ observations per group for multi-group modeling (see Kline, 2005). The small sample is a sizable limitation for multi-group path analysis. Is the sample size (N=130) enough to test the hypothesized models across four groups?  

Missing values are acceptable ranges to use full information maximum likelihood with MLR. It is not clear why the author(s) use multiple imputation (MI) instead of FIML. I recommend providing a rationale for using MI. I recommend providing a table for missing values at item and construct level in supplementary material.

The author(s) indicated that there are no significant associations within the entire group. It is more likely that it is related to the small sample size (There are 4 groups, and the sample size is 130 for the entire group).

Results

Tucker-Lewis index (TLI) was not reported to evaluate the model fit. What are the TLI values for each model? Please add references and explain your rationale for not using TLI in the model evaluation. Please Provide clear information about measurement invariance models (configural, metric, scalar?)  and a table for the nested models used in this study. Why didn’t the author(s) use changes in SRMR to compare nested models? (See Chen, 2007).

Discussion

I recommend providing a brief explanation of the significance and unique aspects of this study at the beginning of the discussion section.
